# Influence of Silver Nanoparticles on the Growth of Ascitic and Solid Ehrlich Adenocarcinoma: Focus on Copper Metabolism

**DOI:** 10.3390/pharmaceutics15041099

**Published:** 2023-03-29

**Authors:** Daria N. Magazenkova, Ekaterina A. Skomorokhova, Mohammad Al Farroukh, Maria S. Zharkova, Zena M. Jassem, Valeria E. Rekina, Olga V. Shamova, Ludmila V. Puchkova, Ekaterina Y. Ilyechova

**Affiliations:** 1Research Center of Advanced Functional Materials and Laser Communication Systems, Institute of Advanced Data Transfer Systems, ITMO University, 197101 St. Petersburg, Russia; 2Institute of Biomedical Systems and Biotechnology, Peter the Great St. Petersburg Polytechnic University, 195251 St. Petersburg, Russia; 3Department of Molecular Genetics, Research Institute of Experimental Medicine, 197376 St. Petersburg, Russia; 4Federal State Budgetary Scientific Institution, Saint Petersburg State University, 199034 St. Petersburg, Russia; 5Department of General Pathology and Pathophysiology, Research Institute of Experimental Medicine, 197376 St. Petersburg, Russia

**Keywords:** copper status, copper metabolism, tumor growth, silver nanoparticles, Ehrlich adenocarcinoma, copper-dependent regulatory factors

## Abstract

The link between copper metabolism and tumor progression motivated us to use copper chelators for suppression of tumor growth. We assume that silver nanoparticles (AgNPs) can be used for lowering bioavailable copper. Our assumption is based on the ability of Ag(I) ions released by AgNPs in biological media and interfere with Cu(I) transport. Intervention of Ag(I) into copper metabolism leads to the replacement of copper by silver in ceruloplasmin and the decrease in bioavailable copper in the bloodstream. To check this assumption, mice with ascitic or solid Ehrlich adenocarcinoma (EAC) were treated with AgNPs using different protocols. Copper status indexes (copper concentration, ceruloplasmin protein level, and oxidase activity) were monitored to assess copper metabolism. The expression of copper-related genes was determined by real-time PCR in the liver and tumors, and copper and silver levels were measured by FAAS. Intraperitoneal AgNPs treatment beginning on the day of tumor inoculation enhanced mice survival, reduced the proliferation of ascitic EAC cells, and suppressed the activity of HIF1α, TNF-α and VEGFa genes. Topical treatment by the AgNPs, which was started together with the implantation of EAC cells in the thigh, also enhanced mice survival, decreased tumor growth, and repressed genes responsible for neovascularization. The advantages of silver-induced copper deficiency over copper chelators are discussed.

## 1. Introduction

The link between the neoplasia, growth, and malignancy of tumors and copper metabolism has been studied for several decades. The results of these studies can be summarized in the following theses.

Inherited disorders in copper metabolism leading to local copper accumulation eventually result in carcinogenesis. One of the examples is Wilson’s disease (WD). In this disease, mutations in copper-transporting ATPase P1 type (Wilson ATPase, ATP7B) lead to the impairment of copper excretion and the accumulation of copper in toxic quantities by hepatocytes, inevitably resulting in the development of hepatocellular carcinoma [1]. This pattern is reproduced in LEC rats, which are a geno- and phenocopy of WD in humans [2] as well as in mice with ATP7B knockout [3]. Mechanistically, copper ions can increase the production of ROS that induce oxidative stress and inflammation, which are intrinsically linked to malignant cell transformation [4].

At the early stages of tumor development, two processes are induced: reprogramming of energy metabolism (Warburg effect) and neovascularization. Both processes are controlled by copper-dependent factors. Accordingly, hypoxia-inducible transcription factor (HIF1α) undergoes hypoxia-mediated stabilization [5], binds copper [6,7], and then interacts with *cis*-elements in more than 100 genes, including genes responsible for stimulation of glycolysis and further development of the Warburg effect [8]. The important participants of the neovascularization process are vascular endothelial growth factor (VEGF) and its receptor (VEGFR), the activity of the latter depending on copper binding [9]. The activity of VEGF is also regulated by copper-dependent HIF1α [10,11]. Moreover, copper ions are required for the formation of nitric oxide, which is also a stimulator of angiogenesis [12].

Copper is a cofactor of lysyl oxidase (LOX), which catalyzes the crosslinking of collagen and elastin in the extracellular matrix; traditionally [13] and paradoxically, it regulates gene transcription, motility/migration, and cell adhesion [14]. Together with lysyl-oxidase-like proteins (LOXL), which also contain copper in their active centers, LOX forms a LOX/LOXL family [15]. In tumors, LOXL regulate multiple signaling networks. Their actions are associated with invasion, metastasis, and angiogenesis. Blocking of copper insertion into LOX/LOXL proteins by ATP7A silencing results in a loss of LOX-dependent mechanisms of metastasis [16]. It has been shown that cancer cells secrete LOX/LOXL enzymes to increase the degree of collagen linearization and rigidity of the tissue, facilitating the spreading of metastases and formation of premetastatic niches in the remote regions [17]. In addition, it has been shown that LOXL factors act as promoters of cell motility/migration through the suppression of E-cadherin gene expression. The activity of LOXL is increased in tumors as compared to the parent organ tissues. LOXL2 and HIF1α reinforce each other in cancer. This is linked to the activation of the PI3K/Akt signaling pathway by LOX with the consequent up-regulation of HIF1α protein synthesis in a manner requiring LOX-mediated hydrogen peroxide production [18,19]. Hence, both proteins act synergistically by promoting the adaptation of the tumor to hypoxia and the stimulation of its progression [14,16,18].

Copper controls the activity of pro- and antiapoptotic systems. Accordingly, an increase in the intracellular copper level, which is regulated by COMMD1 (cytosol Cu(II)-binding protein) [20], causes weakening of the specific interaction between XIAP (the cytoplasmic X-linked Inhibitor of Apoptosis Protein, an E3 ubiquitin ligase) and caspases that induce apoptosis [21,22,23]. It was shown that the content of XIAP increases in some cancers, e.g., breast cancer and renal cell carcinoma [24,25]. The interaction of XIAP with COMMD1 causes the subsequent degradation of COMMD1, which participates in copper excretion. It causes an increase in copper contents in the cells, which can further be used by the growing tumors [22]. It is important that XIAP regulation is disturbed in WD [26], evidencing for the link between copper metabolism and apoptosis.

The mitochondria possess a sophisticated system for copper delivery to the matrix for the maturation of mitochondrial copper-containing enzymes. Moreover, mitochondria have a system for uptake and release of copper from/into cytosol. Accordingly, mitochondria control cellular copper balance [27,28,29]. Copper accumulation in mitochondria causes mitophagy [30] or a copper-dependent form of cell death named cuproptosis, which occurs via copper binding to lipoylated enzymes in the tricarboxylic acid cycle and the loss of Fe–S proteins resulting in protein aggregation, proteotoxic stress, and ultimately cell death [31]. The development of cancers and WD is linked to cuproptosis control [32].

Thus, copper’s connection to cancer is a very hot topic. Recently, the molecular mechanisms responsible for this interplay have been termed cuproplasia (copper-dependent cell proliferation) [33]. The admission of this link stimulated the search for antineoplastic agents that suppress tumor growth by influencing copper status through copper chelation. For now, the search has been running for more than 20 years [33,34,35,36,37,38,39,40,41]. Copper chelators, which can sequestrate copper ions from the cells, are used as candidate drugs. Primarily, it is tetrathiomolybdate (TTM), a drug which has been used for many years to treat WD by reducing copper levels in the livers of WD patients [34]. However, TTM binds dietary copper in the small intestine and blocks its further transport in the organism [36]. Due to this, a constant monitoring of copper levels is needed to prevent the deficiency of vital copper-containing enzymes, such as superoxide dismutase, cytochrome-*c*-oxidase, and ceruloplasmin [42]. In studies of breast cancer cases, it has been shown that TTM facilitates the transition of the tumor to dormant state and prevents relapses [39]. However, in other clinical tests with different tumors, no significant positive effects of TTM were observed [38,43]. Accordingly, the search for other agents, which would be more efficient, safer, and have less side effects, is underway.

Several cases of successful use of silver nanoparticles (AgNPs) for the inhibition of the growth of Ehrlich ascites in vivo and several cancer cell lines in cell cultures have been reported [44,45,46,47,48]. There is no universally accepted concept that would explain the action of AgNPs up to date. Meanwhile, the antitumor effect of AgNPs can be explained by the release of Ag(I) ions from the AgNPs’ surface and the entry of Ag(I) into the Cu(I)-transporting pathways because of the similar coordination properties of Ag(I) and Cu(I) and the inability of the Cu(I)-binding proteins to segregate them [49,50]. Ag(I) ions are transported to the lumen of the Golgi complex, where they efficiently compete with copper for the active sites of maturing ceruloplasmin (Cp), the main copper-containing protein of blood serum (Figure 1). As a result, an oxidase-deficient Ag-Cp that lacks normal copper-transporting function [51,52] circulates in the bloodstream, while copper status indexes (serum copper concentration and Cp oxidase activity) decrease by several times. In this way, the levels of copper available to tumors also decrease.

In the present study, this hypothesis was tested in vivo in ascites and solid Ehrlich adenocarcinoma (EAC) using AgNPs produced by the chemical reduction in AgNO_3_ by hydrazine hydrate with potassium oleate as a capping agent. The paper describes the protocols of treatment of mice with growing ascitic or solid Ehrlich carcinomas (ascitic EAC and solid EAC, respectively) with AgNPs that result in a significant decrease in tumor growth. It was shown that the retardation of tumor growth is associated with an abrupt drop of the copper status indexes of blood serum to practical zero levels. It has been also shown that freely growing EAC cells and the same cells growing in tissue display different patterns of the expression of genes that encode copper-dependent proteins. The possible reasons for the antitumor action of AgNPs and the difference in intracellular copper metabolism in ascites and solid tumor EAC cells are discussed.

## 2. Materials and Methods

### 2.1. Fabrication of AgNPs and Their Physicochemical Characterization

AgNPs were produced chemically by reduction of silver nitrate by hydrazine hydrate in the presence of potassium oleate as a stabilizer. The reagents were purchased from “Reakhim” (Moscow Region, Russia). The size of nanoparticles was controlled by the variation of silver nitrate and potassium oleate concentrations in the reaction volume. The crystalline nature of the synthesized AgNPs was confirmed by UV/vis absorption spectrometric scanning with Shimadzu UV 1800 (Shimadzu, Kyoto, Japan) and energy-dispersive X-ray (EDX) spectroscopy using microscope Jeol JEM-2100F X-ray spectrometer with Bruker XFlash 6-T30. The size and shape of the AgNPs were characterized by transmission electron microscopy (TEM) using Jeol JEM-2100F microscope (Jeol, Tokyo, Japan). Silver concentration was measured by atomic absorption spectrometry (AAS) (ZEEnit 650P spectrometer, Analytik Jena, Germany). In the present paper, the AgNP concentrations are specified as gross molar silver concentration [Ag] (moles of silver per liter of medium), calculated from the AAS measurements. To determine the pool of ionic silver, aliquots of the AgNPs were centrifuged at 30,000× *g* at 4 °C. Silver concentration was measured in supernatant after 1, 2, and 3 h by AAS. During at least one year, the AgNP samples remained stable; aggregates were not formed in it, and no color changes were observed.

### 2.2. Animals

Eight-week-old female Balb/C mice with body weights about 20 g were purchased from the Stesar nursery (Vladimir Region, Russia). The animals were maintained in polycarbonate cages with wood shavings in a temperature-controlled facility (23–25 °C), with a 12:12-h light-dark cycle, 60% humidity with food and water ad libitum. Two weeks before the experiments, the animals were acclimatized to the laboratory environment. Procedures involving animals and their care were conducted according to the guidelines of the Declaration of Helsinki and approved by the Institutional Ethics Committee the Institute of Experimental Medicine (protocol 1/20 of 27 February 2020).

### 2.3. Treatment of Mice with AgNPs and Ehrlich Ascites Carcinoma Cells

Before the injections, the AgNP preparation was suspended in phosphate-buffered saline (PBS: 137 mM NaCl, 2.7 mM KCl, 1.8 mM KH_2_PO_4_, 10 mM Na_2_HPO_4_, pH~7.4, physiological ionic strength). To ensure homogeneity of the AgNP samples, they were treated by ultrasound (37 kHz) in USB2-0.16/37 device (VNIITVCH, St. Petersburg, Russia) for 5 min before to use. The used mode of treatment did not heat the samples and did not affect the biological and physicochemical properties of the AgNPs.

Ehrlich ascites carcinoma (EAC) was sustained in Balb/C female mice by transplanting ~10^7^ cells intraperitoneally (in 0.3 mL) from a mouse with ascites to a healthy mouse every 10 days. For the experiment, on the tenth day after tumor inoculation, 1.0–1.5 mL of ascitic fluid containing EAC cells were aspirated from the abdominal cavity of the mouse with a syringe. Cells were diluted and washed twice with sterile PBS by centrifuging at 300× *g* at 4 °C, the supernatant was discarded, and the precipitated cells were resuspended in the fresh portion of PBS. Then EAC cells’ concentration was adjusted to the desired level for the inoculation of the experimental groups to be used in accordance with specific tasks, which are described at the appropriate places in the section “Results and discussion”.

At the end of experiments, the mice were sedated with diethyl ether vapor and euthanized by cervical dislocation, which was performed by skilled personnel. The blood samples were collected from the ocular vessels, and after clot formation, the serum was separated by centrifugation at 2000× *g* for 10 min at 4 °C. Fecal samples from the upper small intestine and lower intestine were weighed, diluted three times with PBS, suspended, and centrifuged 10,000× *g* for 10 min at 4 °C. The supernatants were collected, and the silver concentration was determined in them. The blood serum and tissue samples were frozen and stored at −80 °C before use.

### 2.4. Gene Expression Analysis

The relative expression of copper-associated genes was measured by quantitative RT-PCR (qPCR). Total RNA was extracted from the liver, ascitic cells and solid tumors by Rizol (Evrogen, Russia). RNA quantity and quality were assessed by a NanoDrop 1000 (Thermo Fisher Scientific, Waltham, MA, USA). First strand cDNA from RNA templates was synthesized using RevertAid First Strand cDNA Synthesis Kit (Thermo Fisher Scientific, Waltham, MA, USA). qPCR was carried out by using qPCRmix-HS SYBR (Evrogen, Russia) with specific primers according to the manufacturer’s instructions. The gene expression analysis was performed using the 2^−ΔΔCt^ method and expressed as fold change relative to the control. The specificity of PCR products was confirmed at the end of PCR by analyzing the DNA melting curve.

### 2.5. Biochemical Methods

Specific activity of ceruloplasmin (Cp) was detected using the assay-in-gel method. Blood serum aliquots (2 μL) were fractioned in non-denaturing 8% polyacrylamide gel (PAG) electrophoresis (PAGE). Gels were stained with *o*-dianisidine to reveal the Cp oxidase activity [53].

For immunoblotting analysis (WB), electrophoresis of serum aliquots (0.1 μL) in 8% PAG with 0.1% SDS according to Laemmli method was performed in Mini-PROTEAN system (“Bio-Rad”, Hercules, CA, USA). Protein markers with molecular masses ranging from 10 to 250 kDa were purchased from “Bio-Rad” (Hercules, CA, USA). After electrophoresis, proteins were transferred from PAG to the Hybond ECL nitrocellulose membrane (GE Healthcare, Chicago, IL, USA) using the same Wet Mini-PROTEAN blotting system (“Bio-Rad”, Hercules, CA, USA). Control of the transfer quality and uniformity was performed by Ponceau S staining. Nitrocellulose membrane was blocked with 5% skimmed milk. Noncommercial rabbit antibodies raised to high purity murine Cp (A610/280 = 0.0469) isolated through its interaction with neomycin, were used as primary antibodies [54]. Horseradish peroxidase-conjugated goat anti-rabbit antibodies (Abcam, UK) were used as the secondary antibodies. Visualization of the immune complexes was performed in ChemiDoc System (“Bio-Rad”, Hercules, CA, USA) using Clarity Western ECL Blotting Substrate (“Bio-Rad”, Hercules, CA, USA). Stained electrophoretic gels and membranes with visualized antigen bands, obtained by WB, were analyzed using ImageJ software (NIH, USA) and expressed as arbitrary units (a.u).

The concentration of the cells in suspension, as well as later in the samples of ascitic fluid obtained from the experimental mice, was determined using a hemocytometer. Cells were preliminary stained with 0.2% trypan blue dye in PBS for 3 min at room temperature; only viable cells (which do not consume the dye) were counted.

Metal concentrations in biological samples were measured using graphite furnace AAS with a Zeeman correction of nonselective absorption using a ZEEnit 650P spectrometer (Analytik Jena GmbH, Germany). Tissue samples for AAS were dissolved in pure HNO_3_. All solutions were prepared in deionized water.

The significance of changes was determined by Mann–Whitney test (StatSoft Statistica 6.0; Tulsa, OK, USA); the changes were considered significant at * *p* < 0.05.

## 3. Results and Discussion

### 3.1. Physicochemical Specification of Fabricated Silver Nanoparticles

Physicochemical characteristics of the fabricated nanoparticles that were used in the present study are summarized in Figure 2. The comparison between element composition of the AgNPs’ aqueous “solution” (Figure 2a) and the substrate (Figure 2b) performed by EDX microanalysis indicates that the sample is practically pure silver. UV/vis spectrometric studies of the AgNPs’ dispersion revealed a broad absorption band with the maximum at approximately 400 nm (Figure 2c), which is characteristic of metallic silver nanoparticles. The solution of the AgNPs had a dark brown color (Figure 2c, inset). No cases of discoloration, particles’ sedimentation, or aggregation of the AgNPs were observed. Atomic concentration of silver in the sample measured by AAS was 3.5 mM. According to the TEM data, the sample contained mostly spherical AgNPs with a median diameter of 30 nm and a small number of nanoparticles with a diameter of 2 nm (Figure 2d). Both types of AgNPs had a crystalline structure: a comparison of the XRD spectra with the standard ones confirmed that the formed AgNPs were nanocrystals, as evidenced by the peaks at 2θ values of 32.3°, 46.3°, 57.5°, and 77.0°, which corresponded to the (111), (200), (220), and (311) planes for silver, respectively (Figure 2e). Centrifugation of the AgNP samples for 3 h led to almost complete sedimentation of silver (Figure 2f).

Accordingly, in the present work we used a stable homogeneous preparation of 30 nm AgNPs free from ionic silver. The fabricated nanoparticles did not differ from the previously used spherical AgNPs with a diameter of 30–70 nm, obtained by the same method [55].

### 3.2. Design of AgNP Treatment of Mice with Growing Ascitic or Solid Ehrlich Adenocarcinomas

This study of the influence of the AgNPs on the survival of mice with growing EAC was designed using the paradigm that the reduction in copper status by the AgNPs can suppress tumor growth [33]. Accordingly, several points had to be addressed: (1) what dose would be preferable, (2) what method of AgNP injection would be the most appropriate and whether it depended on the localization of the tumor, and (3) what periodicity of treatment should be used.

In this study, we used low dose of AgNPs (2 mg silver per 1 kg body mass) that do not disturb hepatic and renal functions (according to concentration of ALT, ACT, bilirubin, and creatinine in blood) during the application for three weeks, as proved by preliminary and previously reported studies [55]. Such doses are neither cytotoxic nor genotoxic [56]. Currently, intraperitoneal (IP), intranasal (IN), oral (PO), or intravenous (IV) applications are used to study the bioactivity of AgNPs. Previously, we tested the listed methods of application for their ability to reduce copper status indexes. It was found that IP and IV modes of treatment produced the strongest and almost similar effects, but IP mode was less invasive and thus preferable for chronic application [57].

One more important question was the periodicity of the treatment that was required to keep the copper status indexes at low levels. To address this question, the experiment of pulse-chase type was applied. The mice were treated with a single intraperitoneal injection of AgNPs. Then silver and copper concentrations in blood serum and liver tissue as well as the Cp oxidase activity and the Cp protein concentration were measured after 5, 10, 15, 30 min and 1, 2, 4, 8, 16, 24, 32 h. Three mice were used for each time point. The data are summarized in Figure 3. In 2 h, copper concentration in blood serum reduced by 25% (Figure 3a). The level of the oxidase activity decreased synchronously (Figure 3b). At the same time, the Cp protein concentration did not change (Figure 3b). The decline of the copper concentration and oxidase activity was accompanied by the initial increase in the silver concentration (Figure 3c). The silver concentration peaked at several hours after the injection and then started to drop, decreasing to small level after 24 h. Additionally, the silver concentration was measured in the upper and lower intestine. Based on data about silver transfer along the intestine, the kinetics of the decrease in silver concentration in blood corresponded to the silver excretion via bile (Figure 3d). These results show that one injection of a small concentration of fabricated AgNPs is enough to reduce copper status indexes, but it is not enough to achieve complete inhibition of oxidase activity. Moreover, the large fraction of silver is excreted within a day, which may potentially lead to the recovery of copper status [58].

To determine the optimal number of injections of the AgNPs for supporting the low level of copper status, silver accumulation in blood and some tissues, the mice were intraperitoneally injected with AgNPs daily for eight days. Three mice were used for each group. In blood serum, a dynamic balance between inflow and excretion of silver was reached in three days and no further accumulation was observed (Figure 3e). At the same time, the oxidase activity decreased to approximately zero in five days (Figure 3e, inset). Three organs were selected for silver accumulation testing: the liver (because absorbed AgNPs are taken up by the liver through the portal vein), the spleen (since the splenic vein is branched from the portal vein), and the lung (as an organ that does not link with the portal vein) (Figure 3f). The silver concentration was very low in the lung when compared to the silver concentration in the liver and spleen. In the liver and spleen, the silver concentration was almost equal. The silver accumulated for three days, and then a balance between silver intake and excretion was achieved. According to these results, in the following IP experiments, the mice were treated with AgNPs daily.

### 3.3. Influence of AgNPs on Survival of Mice with Growing Ascitic or Solid EAC

In this part of this study, survival curves were used as the main estimator of potential antitumor action of the AgNPs in the mice with growing EAC. For that, the following schemes of AgNP treatment have been examined: EAC cells could be transplanted to the mice with already decreased copper status indexes by injections of the AgNPs, and the AgNPs’ treatment could be continued or canceled after the transplantation. Alternatively, tumor inoculation and the AgNPs’ treatment could be started simultaneously (Figure 4). The mice that were inoculated with EAC cells but not treated with AgNPs were used as a control group (the group is not shown in Figure 4). Each group consisted of 18 animals. If EAC cells were transplanted into the mice with low copper status, followed by cancellation of the AgNPs’ administration (Figure 4, group 01), tumors develop aggressively, and it is expressed in a sharp decrease in the survival of the mice. These results correspond to earlier data for the nude mice with growing tumors of human HCT116 cells treated with silver ions [59]. They also correspond to the effects of TTM on tumor growth: it was shown that tumor growth, which was retarded during the TTM treatment, became more aggressive after TTM cancellation [60]. The effect may be explained by the fact that copper-lowering agents (copper chelators or AgNPs) dampen the activity of copper-dependent oncosuppressors, as it was shown that stable transfection of full-length LOX cDNA into a gastric cancer decreased its proliferation [61].

In the mice of group 02, the IP injections of the AgNPs were started 7 days before intraperitoneal or intramuscular EAC cells inoculation, and then were continued after tumor transplantations. Such AgNPs’ treatment did not affect survival of the mice bearing either ascitic or solid tumor compared with the control group of mice (Figure 5a,b). In the mice of group 03, in which the IP treatment with the AgNPs was started simultaneously with IP inoculation of the EAC cells, survival significantly increased (Figure 5c). If the mice were treated with IP injections of the AgNPs simultaneously with intramuscular inoculation of the EAC cells (group 04), the survival of the animals was significantly reduced compared to the control (Figure 5d). However, if intramuscular inoculation of the EAC cells (in the right thigh) and intratumoral (IT) administration of the AgNPs (in the right thigh inside growing tumors) were performed simultaneously (group 05), the survival of the animals was significantly higher than in the control group (Figure 5d).

The number of tumor cells in ascites of the mice treated with IP injections of the AgNPs was lower than in the untreated mice with ascitic EAC (Figure 6a). In animals with solid tumors, IP administration of the AgNPs did not lead to a significant reduction in tumor size, while in the mice with local administration of the AgNPs (group 05), tumor size was significantly reduced (Figure 6b). Accordingly, these results fully support the data about the survival of the mice presented in Figure 5. Thus, it can be assumed that the increase in the survival of the mice in groups 03 and 05 is associated with the suppression of the tumor growth. Moreover, it is possible that IT injection of the AgNPs can be more effective than IP treatment.

### 3.4. Characterization of Copper Status in Mice with Ascitic or Solid EAC Treated with AgNPs

In the second part of this study, we used 8 groups of mice (Table 1). Group 1—the control group of animals without any treatment. In groups 2 and 7, the mice were treated with AgNPs intraperitoneally (IP) or in the right thigh around the potentially growing tumor, respectively. In groups 3 and 5, the mice were inoculated with EAC cells intraperitoneally (IP) or in the right thigh, respectively. In group 4, the mice with ascitic EAC were treated with AgNPs IP from the first day of inoculation during the whole experiment. In group 6 and 8, the mice with solid EAC were treated with AgNPs IP or IT, respectively, from the first day of inoculation during the whole experiment. The mice for biochemical analysis from groups 1–8 were sacrificed on the 14th day of experiment.

One of the common traits of tumor growth both in humans and in animal models is the increase in the Cp level in blood serum [59,62]. This increase generally does not depend on tumor origin or localization and starts to manifest at early stages of tumor growth [59]. However, little is known about tumor-associated changes of copper homeostatic system in the liver, which serves as the source of serum Cp, and determines the main copper status indexes of blood serum: total copper concentration, oxidase activity, and the Cp protein content [63]. It is also unknown whether the type of the tumor (ascitic or solid) affects copper status. These questions were partly addressed in the following experiments. Since groups of the intact mice and the mice treated only with AgNPs (without tumor) are introduced for comparison starting from the current section, for the convenience of the reader, the color and numerical designations of the animal groups are summarized in Table 1 and are used in all following figures.

Compared to the intact control, growth of ascitic as well as solid EAC induced an increase in the copper concentration, oxidase activity and Cp protein level (Figure 7a,c,f, groups 3 and 5 vs. group 1). The assessment of copper status in the mice bearing ascitic EAC treated with AgNPs IP has revealed that copper concentration and oxidase activity, but not the Cp protein concentration, decreased (Figure 7a,c,f, group 4 vs. group 3 *). In the mice treated only with AgNPs IP, oxidase activity, but not the Cp protein concentration, was reduced (Figure 7c,f, group 2 vs. group 1). Hence, the reduction in the Cp-associated oxidase activity during AgNP treatment (Figure 7c, group 4), the suppression of the proliferation of ascitic EAC cells (Figure 6a), and an increased survival of the mice with ascitic EAC treated with AgNPs (Figure 5a) were correlated with each other.

The treatment of the mice bearing solid EAC by IP AgNPs injections resulted in a decrease in the copper concentration and oxidase activity as well as in the immunoreactive Cp level (Figure 7a,d,g, group 6 vs. group 5). Compared with the IP administration, the reference topical injection of AgNPs into the right thigh of the intact mice did not cause changes in copper status indexes (Figure 7a,e,h, group 7 vs. group 1). The treatment of solid EAC with AgNPs using IT injection resulted in an increase in the Cp protein concentration that was not proportional to the oxidase activity and copper concentration (Figure 7a,e,h, group 8 vs. group 7). These results show that IT treatment of tumors with AgNPs increased the level of the Cp protein but not the oxidase activity (Figure 7h, group 8), reduced the rate of tumor growth (Figure 6b), and increased the survival of the mice (Figure 5b). The data suggest that solid tumor development may be dependent on Cp-associated copper and support a concept of the important role of holo-Cp in tumor development.

The silver concentration in the blood serum of the mice treated with AgNPs IP was 2 times higher than in the mice with the IT AgNP injections (Figure 7b, group 2 vs. group 7). Perhaps this is due to the fact that the AgNPs were used in 10 times less concentration for IT treatment. In both ascitic and solid EAC, the serum silver concentration significantly decreased during the IP treatment with the AgNPs (Figure 7b, groups 4 and 6 vs. group 2). In contrast to this, the IT injection of the AgNPs into solid EAC did not change the silver concentration (Figure 7b, group 8 vs. group 7). The decrease in the oxidase activity did not correlate with the level of immunoreactive Cp determined by immunoblotting (Figure 7f–h). This indicates that a loss of copper ions associated with the Cp was observed.

The growth of ascitic EAC with or without the treatment of the AgNPs did not cause a notable change in the copper concentration in the liver (Figure 8a, groups 3 and 4 vs. 1 and 2). By contrast, copper was elevated in the solid EAC mice treated with the AgNPs IP and IT injection (Figure 8a, groups 6 and 8 vs. 5 and 7). It is possible that silver was slowly accumulated in the liver when the AgNPs were topically applied because a 10 times smaller amount of solution was used for the IT injections (Figure 8b, group 7 vs. group 2).

The measurement of the copper and silver concentrations in the ascitic EAC revealed that the IP injections of the AgNPs did not significantly change either the copper or the silver content per cell (Figure 9a, group 4 vs. group 3). In the solid EAC, the IP and IT injections of the AgNPs decreased the copper concentration (Figure 9b, groups 6 and 8, respectively, vs. group 5). Interestingly, the silver concentration in these tumors was very low: 40 times lower than in the liver (Figure 9b vs. Figure 8b), even though both types of EAC cells were in contact with the AgNPs.

### 3.5. Influence of EAC Tumors on Expression of Genes Encoding Copper Homeostasis Proteinsin the Liver

To characterize the intracellular copper status in the liver cells of the mice with transplanted EAC tumors and treated with the AgNPs, we measured the concentrations of the mature transcripts of the key genes responsible for copper distribution in the liver. We selected genes encoding copper importers (DMT1, CTR1, CTR2), cytosolic Cu(I)-chaperones (CCS, ATOX1, COX17), copper transporter from the cytosol to the lumen of the Golgi apparatus (ATP7B), cuproenzymes (Cp, SOD1), extracellular (prion protein, PRNP), and intracellular copper-binding proteins (MT1, metallothionein, which specifically binds copper). The data presented in Figure 10a indicate that the growth of ascites EAC stimulated the expression of *DMT1, ATOX1, Cp* and *MT1* genes in the liver. The changes in the activity of the other studied genes were not significant (data are presented in Appendix A). In the mice with solid EAC tumors, the activities of the same genes were enhanced, except for *ATOX1*. The increase in the level of *ATOX1* mRNA was not significant (Figure 10b). It is interesting that in the mice with solid EAC, the expression of *DMT1* and *Cp* genes in the liver was stronger than in the mice with ascitic EAC. Treatment with the AgNPs did not notably affect *DMT1* and *Cp* gene expression in the liver of the mice with any form of EAC tumor. Their pattern for *MT1* gene was an exception, its expression further increased in the mice treated with AgNPs as compared to its already elevated level in the untreated tumor-bearing mice, and exceeding its level in control group by a factor of 8 (Figure 10a,b).

While the activity of the CTR1 and ATP7B genes, which encode participants of the route that provides Cp metalation: CTR1→ATOX1→ATP7B→Cp, did not change (Figure 10 and Appendix A), the level of the oxidase Cp in the serum of the mice growing tumors increased (Figure 7), and the concentration of copper in the liver of the mice growing solid EAS increased notably (Figure 8). Perhaps this is due to an increase in the expression rate of the *DMT1* gene (Figure 10 and Appendix A). DMT1 transports copper ions in Cu(II) state, in which they are primarily present in the bloodstream. Accordingly, reduction of Cu(II) to Cu(I), which is necessary for CTR1 transport, is not required for DMT1. This speculation is supported by an increase in the expression of the *MT1* gene (Figure 10). It can be assumed that copper ions enter the cells in Cu(II) state via DMT1 into the glutathione↔MT1 redox cycle.

The data presented in Figure 7, Figure 8 and Figure 10 demonstrate the involvement of copper metabolism in tumor development. In the liver of the mice growing tumors, the *Cp* gene and the genes involved in its loading with copper ions are activated, and copper concentration increases. AgNPs generally do not affect the activity of these genes in the liver. They cause only a decrease in the quasi-stationary level of the oxidase Cp in blood serum (Figure 7).

### 3.6. Changes of Copper-Related Genes Expression in Ascitic and Solid EAC Tumors Induced by AgNP Treatment

In this study, we used Ehrlich adenocarcinoma, which originates from mouse mammary gland tumors, as was noted above. There are no specific data on the expression of the genes associated with copper metabolism for this tumor. It is difficult to extrapolate the data for the original tissue, as the pattern of copper-related gene expression in the mammary glands is highly affected by the physiological state of this organ (proliferation, pregnancy, lactation, and involution) [64]. Therefore, comparison is only possible between a tumor from untreated organisms and a tumor from organisms treated with AgNPs. Decline in the activity of genes coding copper-dependent factors (HIF1α, TNF-α and VEGFa), copper transporter DMT1, and GPI-Cp splice isoform of Cp-mRNA were observed in ascites tumors of the mice that received the AgNP treatment (Figure 11a). At the same time, the LOX-mRNA concentration significantly increased.

In contrast to this, in the mice with solid EAC treated by local application of AgNPs, the activities of *VEGFa* and *VEGFR1* were decreased by the treatment, while the activities of *HIF1α, TNF-α* and *LOX* genes were increased (Figure 11b).

The levels of both splice isoforms of Cp-mRNA were also increased in the mice treated with AgNPs. The expression of genes coding for the copper importers DMT1 and CTR1 in the tumors did not significantly change (Figure 11b).

The profiles of copper-related gene expression in ascitic EAC and solid EAC are compared to each other are in Figure 12. These data help to understand better the results presented in Figure 11. The differences in these profiles are relatively small: genes encoding copper transporters (CTR1 and DMT1), intracellular (SOD1), and secretory copper enzymes (Cp and LOX) are expressed by both types of EAC similarly. In ascitic EAC, expression of copper-mediated signaling genes (*TNF-α* and *HIF1α*) were increased. At the same time, in solid EAC, another gene taking part in copper-mediated signaling (*COMMD1*) was increased.

Moreover, the level of expression of VEGFR and VEGFR2 that is responsible for neovascularization was higher in solid EAC. Although these differences are significant, they do not exceed 30%; therefore, it can be assumed that the level of expression of the copper-related genes selected for this study in both types of EAS is similar. Therefore, the data in Figure 11 are significant.

Summing up the obtained data, we want to emphasize that the genes associated with copper metabolism display different expression patterns in ascitic and solid tumors of the same cellular origin (Figure 11).

When planning our experiments, we assumed that ascitic EAC, which consists of individual intraperitoneally suspended cells, does not experience hypoxia. Hence, it was hypothesized that the suppression of ascitic EAC growth would be unrelated or weakly related to the expression of copper-dependent factors. However, the AgNPs suppressed ascitic EAC cell proliferation through decreasing of expression of the genes responsible for copper-dependent signaling and neovascularization. In solid EAC, IT injection of the AgNPs repressed the genes responsible for neovascularization (Figure 11b). It was correlated with the regression of tumor growth. At the same time, we did not observe a profound correlation between the copper-related genes and the suppression of tumor growth, as it was seen for the copper status index. It is possible that there is no direct link between copper metabolism in cancer cells and copper homeostasis in blood serum; however, it is also possible that other approaches are needed to examine it.

## 4. Conclusions

All tumors of similar origin, irrespective of the difference in the susceptibility or resistance to the treatment and prognosis of development, have common characteristics that distinguish them from normal tissues, not to mention tumor type-specific traits such as the variability of the metabolic profiles and energy metabolism [65]. Tumors are characterized by loss of growth-inhibitory signals, hyperactivation of signaling cascades, uncontrolled proliferation, suppression of apoptosis, induction of angiogenesis, nonstop replication, and stimulation of invasion and metastasis processes. Of all these traits, only nonstop replication has no obvious links to copper metabolism, if the activity of [Fe-S]-cluster proteins that are indispensable for replication is not taken to account. All the other tumor-specific features are connected to copper in various ways [5,7,9,11,16,17,20,21,28,29,31,32,33].

The system that ensures the copper turnover in the body comprises an intracellular system that provides the synthesis of cuproenzymes, including secretory ones (Figure 1), and an extracellular system that controls the copper balance in the whole organism [66]. A key player of extracellular copper metabolism is Cp, which indicates the current state of copper homeostasis and corresponds to the physiological requirements of various organs with copper [63,67,68,69,70]. Cp, a representative example of a “moonlighting” protein, is the major extracellular copper-containing protein in vertebrates [71,72]. Several facts prove that Cp is a copper transporter for nonhepatic organs coupled to CTR1 importer. First, in addition to six non-exchangeable copper atoms in the catalytic centers [73], Cp binds at least two labile copper atoms that can be transferred to cells [74]. Second, regions of the Cp molecules that bind labile copper can specifically associate with extracellular copper-binding domain of CTR1 protein according to molecular modeling [75]. Third, Cp can provide copper to cells in cell culture [76]. Fourth, Cp can serve as a copper donor for CTR1 in vivo [77].

This study follows the strategy of suppressing tumors with copper chelators [33] and confirms that copper associated with Cp supports the neoplastic growth. The replacement of this copper with silver atoms, which are metabolically included in the Cp during its metalation in the Golgi apparatus [51,52], leads to the suppression of tumor growth. This is manifested by the data on in vivo treatment with AgNPs, which leads to retardation of tumor growth and increases viability (Figure 5 and Figure 6). Since long-term copper deficiency does not disturb the intracellular balance of copper in nonmalignant tissues, it is safe for normal cells. At the same time, AgNPs, which remove only copper from its path to the tumor, are even more beneficial if compared to TTM and other broad-acting chelators due to their narrowly targeted and highly specific effect on copper metabolism [49,50]. It is possible that the proposed approach may complement the main nonsurgical anticancer strategies: chemotherapy and radiation therapy.

## Figures and Tables

**Figure 1 pharmaceutics-15-01099-f001:**
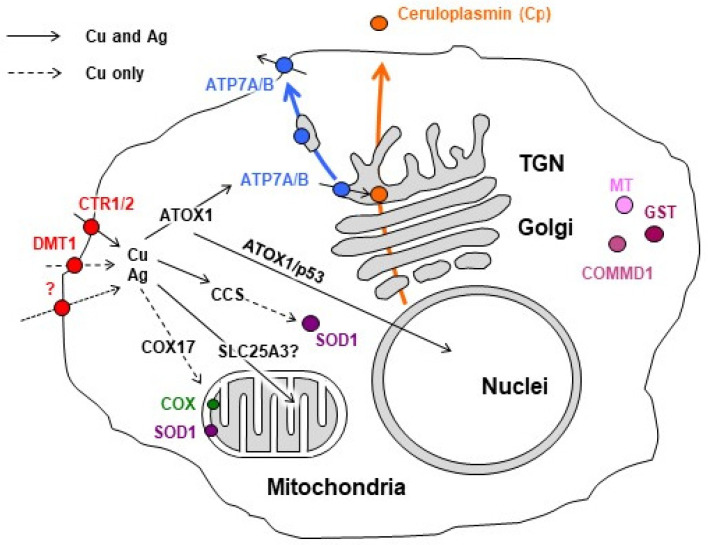
Scheme of copper and silver distribution in a mammalian hepatocyte. Copper is taken up via copper transporter 1 (CTR1), divalent metal transporter 1 (DMT1), or the putative transporter (all depicted as red circles). After being imported into the cell, the copper is transferred to chaperones antioxidant protein 1 (ATOX1), copper chaperone (CCS), and cytochrome-*c*-oxidase (COX17), which ferry it (black arrows) to both copper-transporting ATPase (ATP7A/B, blue) in the Golgi, to Cu,Zn-superoxide dismutase (SOD1, magenta) in the cytosol, and to cytochrome-*c*-oxidase (COX, green) in the mitochondria. Mitochondrial phosphate carrier protein (SLC25A3) transfers copper into the matrix. In the Golgi, ATP7A/B load Cu on newly synthesized cuproenzymes such as Ceruloplasmin (Cp, orange circle), which transport it along the biosynthetic pathway (orange arrow). A significant increase in intracellular Cu induces the export of ATP7A/B (blue arrow) toward the post-Golgi compartments (TGN) and plasma membrane, where it drives the excretion of excessive Cu from the cell. Copper Metabolism MURR1 domain protein 1 (COMMD1) protein is involved in copper transport and protein trafficking/degradation. Excessive Cu could be also bind by cysteine-rich peptides Metallothioneins (MTs, pink)—Silver uses similar copper-transporting routes (solid black arrows). However, several copper-transporting pathways cannot be invaded by silver (dashed black arrows). Scheme is modified from our article [49].

**Figure 2 pharmaceutics-15-01099-f002:**
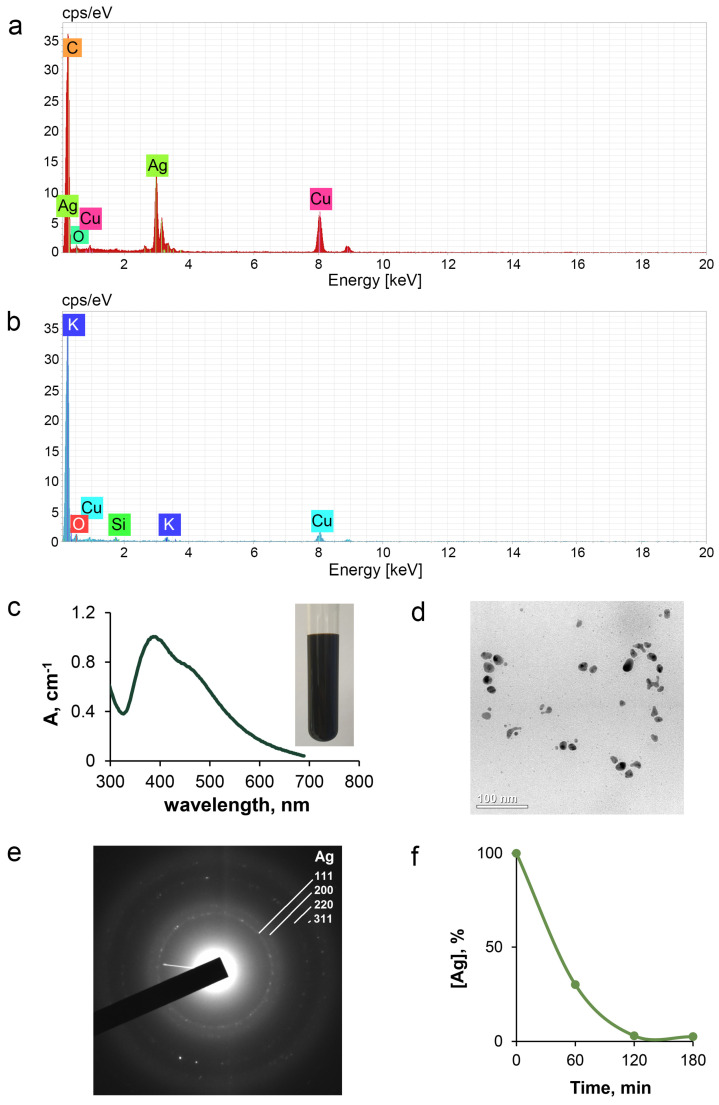
Physicochemical characteristics of fabricated AgNPs. Energy-dispersive X-ray spectrum of (**a**) AgNPs and (**b**) substrate film; (**c**) UV/vis absorption spectra, inset: AgNPs solution; (**d**) Images of the AgNPs obtained by TEM; (**e**) Electron diffraction pattern of the AgNPs; (**f**) Curve of time-dependent sedimentation of nanoparticles. X-axis: centrifuge time, minutes; Y-axis: % of initial silver concentration in the supernatant suspension.

**Figure 3 pharmaceutics-15-01099-f003:**
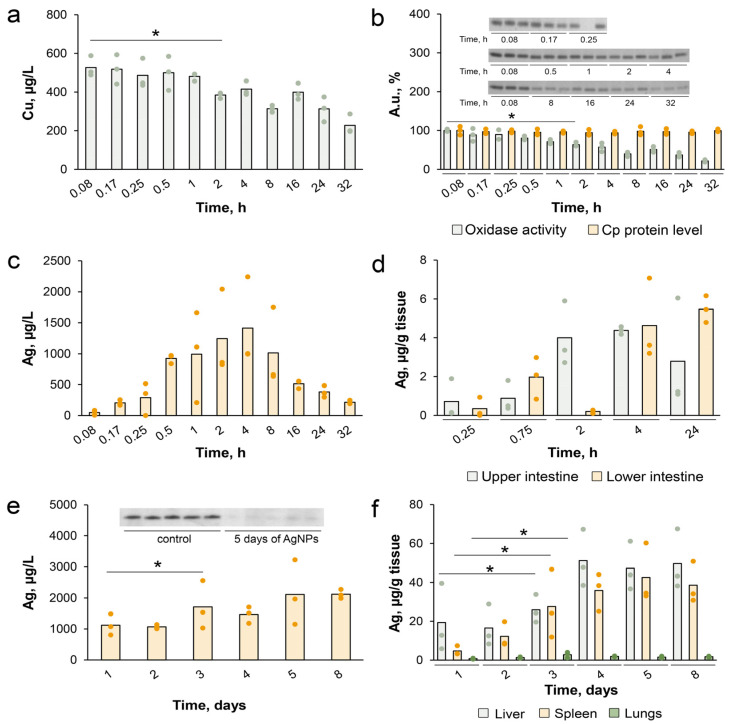
Dynamics of silver and copper distribution in blood serum and liver tissue of mice treated with a single AgNPs injection. (**a**) Concentration of copper in blood serum; (**b**) Serum oxidase activity (green points; sample pictures of in-gel assessment of oxidase activity, which was used to build the bar chart are given above the chart), and Cp protein concentration in the same samples (orange points); (**c**) Concentration of silver in blood serum; (**d**) Silver excretion through the intestine; Dynamics of silver accumulation in (**e**) blood serum and (**f**) organ tissues during 8 days of daily AgNPs injections. The data are presented as mean value (bar) and individual experimental samples (points), *n* = 3. * *p* < 0.05.

**Figure 4 pharmaceutics-15-01099-f004:**
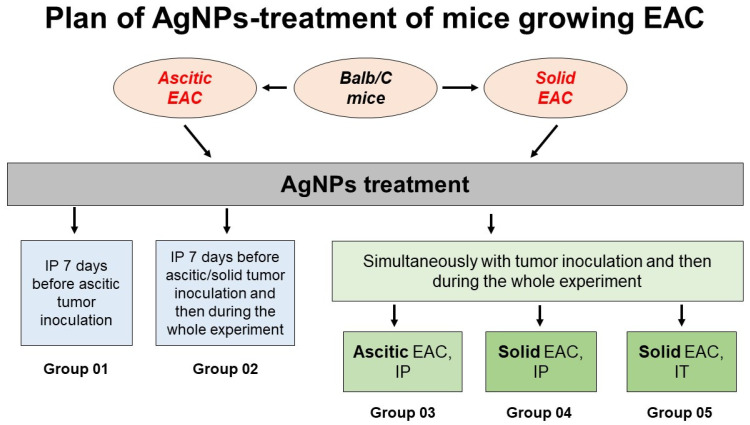
Plan of AgNP treatment of mice growing EAC tumors and numbering of experimental groups. Group 02 includes two subgroups: mice bearing ascitic EAC and solid EAC. In groups 01 and 03, 1 × 10^6^ EAC cells were injected intraperitoneally (IP); in groups 04 and 05, 1 × 10^6^ EAC cells were injected in the right thigh. Mice of 01–04 groups were treated with AgNPs intraperitoneally (100 µL of 3.5 mM AgNPs’ solution per mice). Mice of group 05 were treated with AgNPs inside growing tumors (10 µL of 3.5 mM AgNPs’ solution per mice). Each group consisted of 18 animals.

**Figure 5 pharmaceutics-15-01099-f005:**
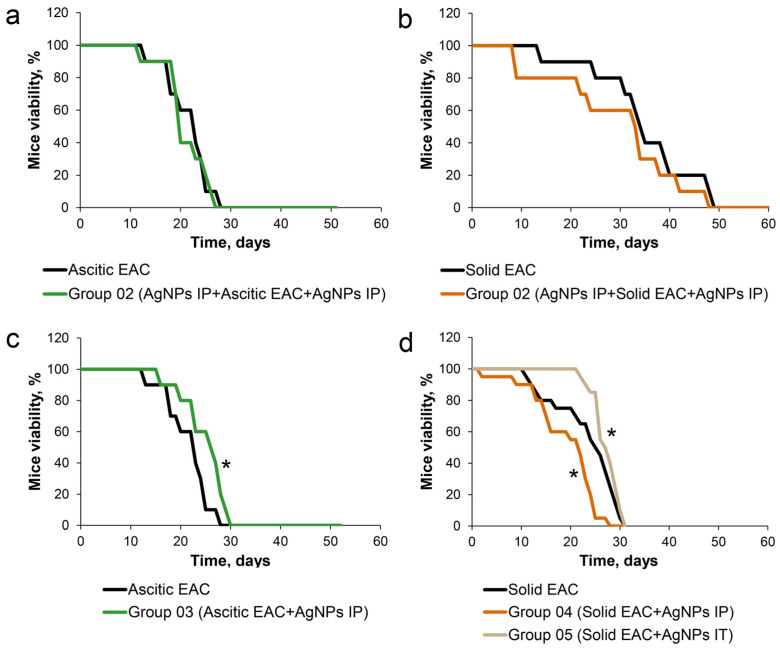
Survival curves of mice bearing ascitic EAC (**a**,**c**) or solid EAC (**b**,**d**) treated with AgNPs. (**a**) Mice received AgNPs IP prior to IP inoculation of EAC cells; treatment with nanoparticles continued after tumor injection till the end of the experiment (mice of group 02 with ascitic EAC in Figure 4). (**b**) Mice received AgNPs IP prior to IT inoculation with EAC cells, treatment with nanoparticles continued after tumor injection until the end of the experiment (mice of group 02 with solid EAC in Figure 4). (**c**): Mice received AgNPs IP simultaneously with inoculation of EAC cells began to receive IP AgNPs; (**d**) Mice received AgNPs IP (brown) or IT (beige) simultaneously with EAC cell inoculation. IP inoculation: 1 × 10^6^ EAC cells were injected intraperitoneally; IT inoculation: 1 × 10^6^ EAC cells were injected in the right thigh. IP treatment with AgNPs: intraperitoneally 100 µL of 3.5 mM AgNPs’ solution per mice; IT treatment with AgNPs: inside growing tumors 10 µL of 3.5 mM AgNPs’ solution per mice. Observation was carried out until all the animals died (*n* = 18). *: *p* < 0.5.

**Figure 6 pharmaceutics-15-01099-f006:**
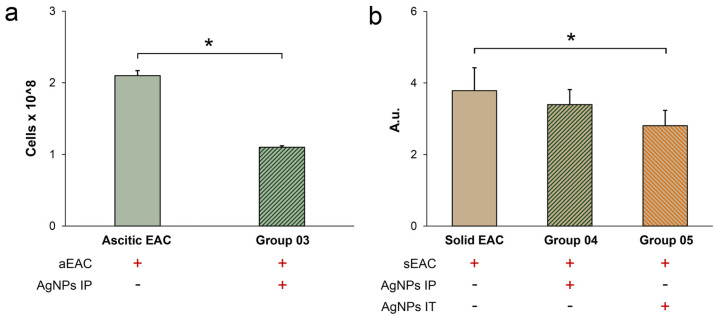
AgNP treatment suppress a growth of ascitic EAC (**a**) and solid EAC (**b**). Y-axis: (**a**) Cell count; (**b**) Relative units. The numbering of animal groups corresponds to the scheme in Figure 4. *: *p* < 0.05.

**Figure 7 pharmaceutics-15-01099-f007:**
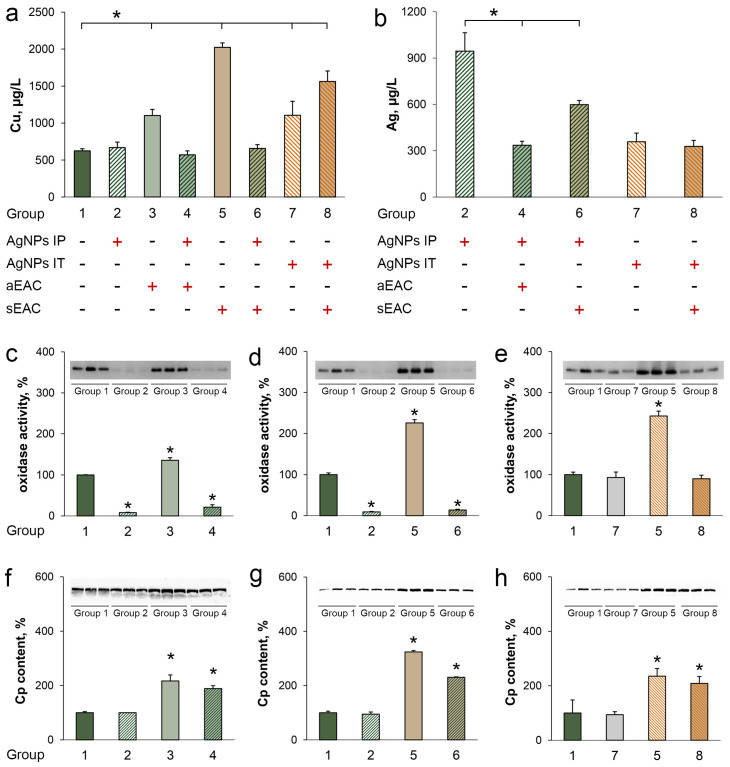
Influence of tumor growth and AgNP treatment on serum copper status indexes. The group numbering is given in Table 1. (**a**) Copper concentration; (**b**) Silver concentration; (**c**–**e**) Oxidase activity, insets-oxidase activity tested in gel with *o*-dianisidine; (**f**–**h**) Cp protein concentration measured by WB, insets-immunoblotting with antibodies to murine Cp. *: *p* < 0.5 compared to group 1 (**a**,**c**–**h**) and 2 (**b**).

**Figure 8 pharmaceutics-15-01099-f008:**
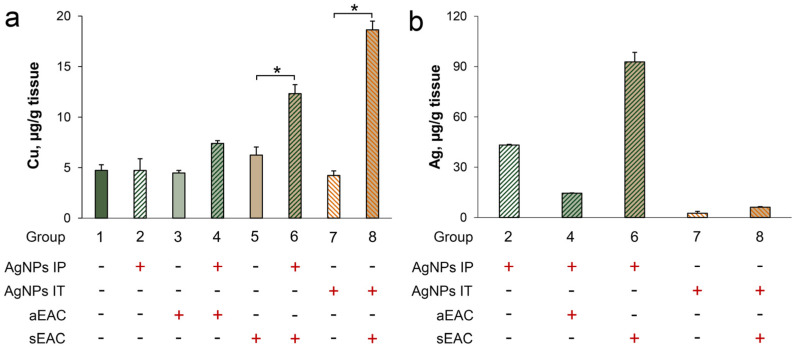
Copper (**a**) and silver (**b**) concentration in the liver of experimental mice. (**a**) Ascitic EAC (1–4) and solid EAC (5–8); (**b**) Ascitic EAC (2 and 4) and solid EAC (6 − IP injection; 7 and 8 − IT injection). *: *p* < 0.5. Numbering of groups is the same as in Table 1.

**Figure 9 pharmaceutics-15-01099-f009:**
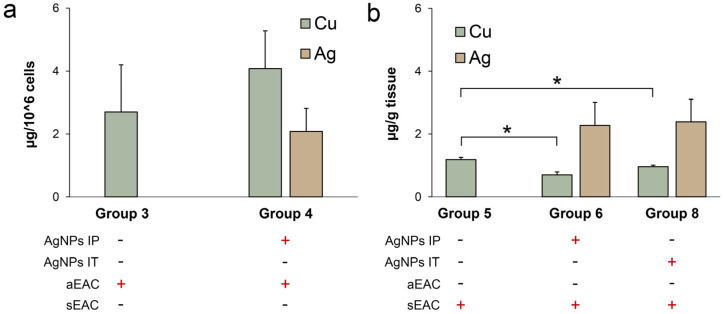
Copper (green bars) and silver (brown bars) concentration in ascitic (**a**) and solid (**b**) EAC. Numbering of groups is the same as in Table 1. *: *p* < 0.05 compared to group 5.

**Figure 10 pharmaceutics-15-01099-f010:**
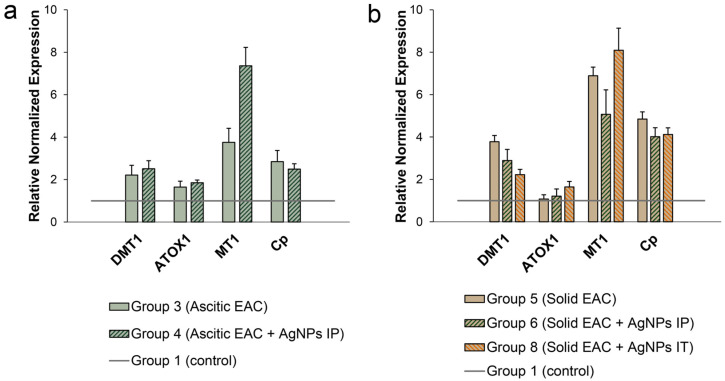
Expression level of the genes responsible for serum copper status in the liver of mice bearing EAC. (**a**) Mice with growing ascitic EAC treated with AgNPs IP; (**b**) mice bearing solid EAC treated with AgNPs IP or IT.

**Figure 11 pharmaceutics-15-01099-f011:**
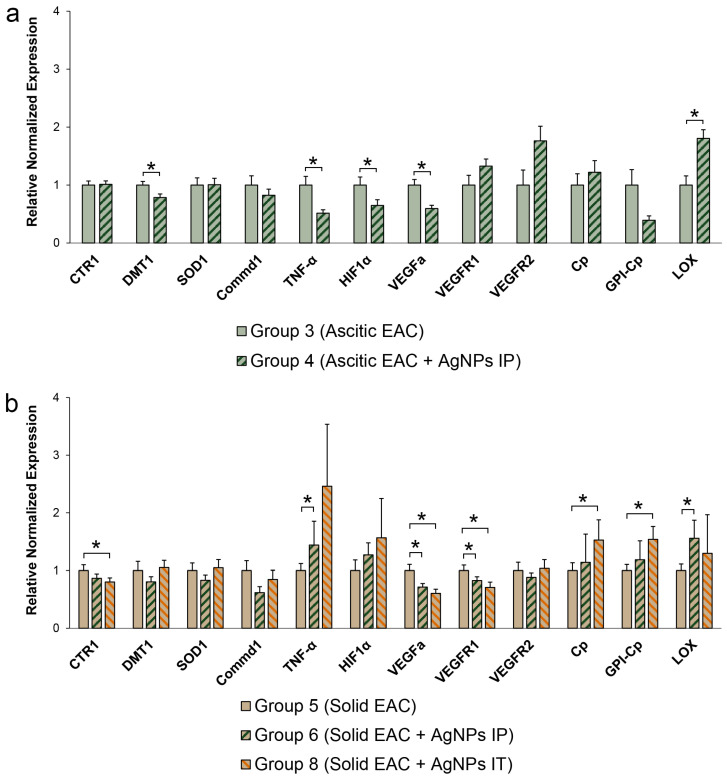
Copper-related genes profile expression in ascitic EAC (**a**) and solid EAC (**b**) treated with AgNPs. Numbering is the same as in Table 1. *: *p* < 0.05 compared to group 3 (**a**) or group 5 (**b**).

**Figure 12 pharmaceutics-15-01099-f012:**
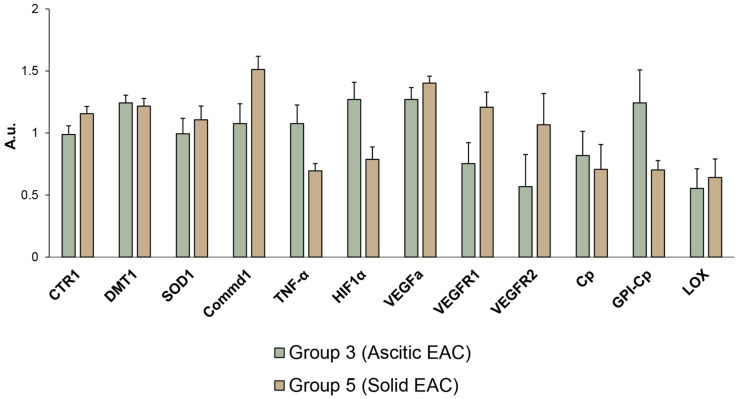
Expression level of the copper-related genes in ascitic EAC and solid EAC.

**Table 1 pharmaceutics-15-01099-t001:** Groups of mice formed to study AgNPs effect on copper status and copper-related genes expression in mice bearing ascitic or solid tumors.

	AgNPs IP	AgNPs IT	aEAC	sEAC
**Group 1** 	**−**	**−**	**−**	**−**
**Group 2** 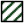	** + **	**−**	**−**	**−**
**Group 3** 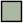	**−**	**−**	** + **	**−**
**Group 4** 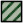	** + **	**−**	** + **	**−**
**Group 5** 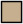	**−**	**−**	**−**	** + **
**Group 6** 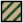	** + **	**−**	**−**	** + **
**Group 7** 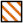	**−**	** + **	**−**	**−**
**Group 8** 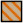	**−**	** + **	**−**	** + **

## Data Availability

Not applicable.

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
