# Peer review of "Influence of Silver Nanoparticles on the Growth of Ascitic and Solid Ehrlich Adenocarcinoma: Focus on Copper Metabolism"

_pharmaceutics, 2023, doi:10.3390/pharmaceutics15041099_

Round 1

Reviewer 1 Report

Despite the interesting topic discussed in the proposed article, there are some drawbacks.

In the materials and methods section, it would also be desirable to indicate the total number of animals used.

The number of samples (n=3) shown in Figure 3 seems too small to objectively assess the effect, and the results in parts d,e,f do not convince of an adequate and correct analytical approach, since the presented mean value with dispersion (mean ± S.D ) contradict the basic postulates of biostatistics, which states that in a normal distribution, 68% of data should be within 1SD, 95% of data should be within 2SD, and 99.7% should be within 3SD. By modeling presented results, randomly generating mean data with SD represented by the orange bar (time 0.25) in Figure 3d gives us negative values, which is a complete alogism and nonsense. These results should be revised.

A thorough revision of the manuscript and correction of all these deficiencies is recommended.

The scientific value of the proposed manuscript would be better if the fundamental error related to the non-observance of one of the basic rules of statistical analysis on the distribution of data and the choice of the correct method for determining the average feature, otherwise the presented results are misleading to the reader and reduce the value of this research work.

Author Response

Reply to the Review Report (Reviewer 1)

Despite the interesting topic discussed in the proposed article, there are some drawbacks.

Dear reviewer, we are very grateful to you for your agreement to review our manuscript, positive evaluation of our work and for your comments. We consider them very useful for our paper and fully take them to account.

In the materials and methods section, it would also be desirable to indicate the total number of animals used.

→        We added an information about number of animals that was used in each experiment in Material and Methods and Results sections. Total number of animals was about 300. All changes are highlighted in blue.

The number of samples (n=3) shown in Figure 3 seems too small to objectively assess the effect, and the results in parts d,e,f do not convince of an adequate and correct analytical approach, since the presented mean value with dispersion (mean ± S.D ) contradict the basic postulates of biostatistics, which states that in a normal distribution, 68% of data should be within 1SD, 95% of data should be within 2SD, and 99.7% should be within 3SD. By modeling presented results, randomly generating mean data with SD represented by the orange bar (time 0.25) in Figure 3d gives us negative values, which is a complete alogism and nonsense. These results should be revised.

→        We used 3 independent mice for each point in experiment of pulse-chase type (33 mice in total) because this is allowable number of animals in the group that it is sufficient for experiments of this type and its statistical analysis. Increasing animals in each would not have led to a qualitative change in the results. At the same time it would not correspond to the International 3R principle of experiments on laboratory animals. According your recommendations we have changed Figure 3 that is significantly improved the quality of our data presentation.

A thorough revision of the manuscript and correction of all these deficiencies is recommended.

→        We have tried to very carefully take into account all the comments that were made and recalculated the statistical analysis.

The scientific value of the proposed manuscript would be better if the fundamental error related to the non-observance of one of the basic rules of statistical analysis on the distribution of data and the choice of the correct method for determining the average feature, otherwise the presented results are misleading to the reader and reduce the value of this research work.

→        We re-performed all the statistical analysis with Mann-Whitney test for small groups without normal distribution.

Reviewer 2 Report

The authors in this manuscript assumed that silver nanoparticles (AgNPs) can be used to lower copper bioavailability in blood stream due to its chelator effect. They investigated this in vivo by injecting ascites and solid Ehrlich adenocarcinoma (EAC) mice with AgNPs. Many experiments were done to check this hypothesis like copper index status to assess the copper metabolism, copper and silver levels and gene expression of copper related genes with RT-PCR. I think the manuscript can be accepted after the authours will make the manuscript more “reader-friendly”.

-          The legend for many figures were not clear and a lot of details should be added to make it more informative. Moreover, the label in the figures and its legend lacking constancy and adherence for instance you can find (A or B) in the figure and suddenly (a and b) in its legend which will contrast the reader. In addition to that, there is some results found in the chart with no experimental proof for example, in figure 3 (B) the gel for serum oxidized activity there is no 0.25, 0.17 time/ h although the chart had it. In particular, the authours should explain in each legend what each group of the animals corresponds to. It was annoying for me to go to the description every time and check this information. The legends should be clear without text.

-          The silver nanoparticles (AgNPs) preparation process need more details and the stability of the nanoparticle was not measured (the uv-vis can be used to test stability after many hours). The size of the nanoparticles should be measured in triplicate and presented as average with SD. The concentration of silver should be measured and presented similarly. Moreover, for color changing and sedimentation test the authours should specify the time instead of using ‘long time’ as a proof of no difference.

-          Regarding the in vivo experiments for AgNPs Influencing the survival of mice with growing ascitic or solid EAC, there are some points need to be clarified for instance: the groups comparison criteria was not identified and the justification for using intratumoral (IT) injection in group (5) while they mentioned earlier the reasons for selecting (IP) intraperitoneal injections in this experiment. also, why did the authours compare ascitic EAC to group (2) which is solid EAC figure 5(C)?

-          The animals grouping for copper status characterization was also not clear and the abbreviations in the table were not defined.

-          The conclusion was extremely long and most of it about copper and its Cp-association while only the last part was about AgNPs, which may unconvince the reader about their findings.

-          The font on all the figures should be larger, it is difficult to read. Additionally, figures 2a and 2b are too small and it is impossible to see anything there.

-          I suggest to remove “first, second, third, fourth, fifth” in the introduction

Author Response

Reply to the Review Report (Reviewer 2)

The authors in this manuscript assumed that silver nanoparticles (AgNPs) can be used to lower copper bioavailability in blood stream due to its chelator effect. They investigated this in vivo by injecting ascites and solid Ehrlich adenocarcinoma (EAC) mice with AgNPs. Many experiments were done to check this hypothesis like copper index status to assess the copper metabolism, copper and silver levels and gene expression of copper related genes with RT-PCR. I think the manuscript can be accepted after the authours will make the manuscript more “reader-friendly”.

Dear reviewer, we are very grateful to you for your agreement to review our manuscript, positive evaluation of our work and for your comments. We consider them very useful for our paper and fully take them to account.

-          The legend for many figures were not clear and a lot of details should be added to make it more informative. Moreover, the label in the figures and its legend lacking constancy and adherence for instance you can find (A or B) in the figure and suddenly (a and b) in its legend which will contrast the reader. In addition to that, there is some results found in the chart with no experimental proof for example, in figure 3 (B) the gel for serum oxidized activity there is no 0.25, 0.17 time/ h although the chart had it. In particular, the authours should explain in each legend what each group of the animals corresponds to. It was annoying for me to go to the description every time and check this information. The legends should be clear without text.

→        We have tried to very carefully take into account all the comments and rewrite figure legends to make them more clear and informative. Legends to figures 3-5 are expanded with information that allows readers not to consult with the text. Groups’ description were added graphically on Figures 6-12. Letters in all figures were changed from uppercase to lowercase. Information about an oxidase activity for 0.17 and 0.25h was added on Figure 3b. All changes are highlighted in blue.

-          The silver nanoparticles (AgNPs) preparation process need more details and the stability of the nanoparticle was not measured (the uv-vis can be used to test stability after many hours). The size of the nanoparticles should be measured in triplicate and presented as average with SD. The concentration of silver should be measured and presented similarly. Moreover, for color changing and sedimentation test the authours should specify the time instead of using ‘long time’ as a proof of no difference.

→        We wrote extended description of AgNPs synthesis and physicochemical characteristics in our previous articles (https://doi.org/10.2147/IJN.S117745; http://doi.org/10.2147/NSA.S287658; https://doi.org/10.1039/D2EN00402J) and have added some more information in this Manuscript (Material and Methods section and Figure 2), however, some details are patent valuable. The size of nanoparticles definitely was measured in several replicates as well as silver concentration (Figure 1a and b in this response). We have written more accurate information about the color of AgNPs solution. According to our observations AgNPs samples did not change their color and there was no sedimentation or aggregation of AgNPs for at least two years (Figure 1c). Here we show you a sample that was fabricated in 2019 and still has the same UV/vis absorption spectra. AgNPs used in this study was fabricated in autumn of 2022 year by the same protocol.

-          Regarding the in vivo experiments for AgNPs Influencing the survival of mice with growing ascitic or solid EAC, there are some points need to be clarified for instance: the groups comparison criteria was not identified and the justification for using intratumoral (IT) injection in group (5) while they mentioned earlier the reasons for selecting (IP) intraperitoneal injections in this experiment. also, why did the authours compare ascitic EAC to group (2) which is solid EAC figure 5(C)?

→        We have added the explanation in section 3.3. We suppose that intratumoral injection of AgNPs may increase its local concentrations, improve the therapeutic index of this treatment and decrease their accumulation by tissues (for example, by the liver).

-          The animals grouping for copper status characterization was also not clear and the abbreviations in the table were not defined.

→        We have changed the description of groups in table, figures, and figures’ legends.

-          The conclusion was extremely long and most of it about copper and its Cp-association while only the last part was about AgNPs, which may unconvince the reader about their findings.

→        According to your recommendations we have completely revised the ‘Conclusion’ section. It was shorten. Also, we tried to prioritize the balance between details of copper metabolism and AgNPs more carefully in order to convince the reader about our findings.

-          The font on all the figures should be larger, it is difficult to read. Additionally, figures 2a and 2b are too small and it is impossible to see anything there.

→        We have changed all the figures and believe that they are easier to understand now.

-          I suggest to remove “first, second, third, fourth, fifth” in the introduction

→        We removed them from introduction.

Round 2

Reviewer 1 Report

The manuscript has now been greatly improved, however the highest grade of evaluation for the manuscript as potential high-level publication still cannot be applied.